# Blockchain-Based Access Control Techniques for IoT Applications

**Sarra Namane** [1] **and Imed Ben Dhaou** [2,3,4,*]

1   Networks and Systems Laboratory, Department of Computer Science, Badji Mokhtar University, Annaba 23000, Algeria; naamanesara2005@yahoo.fr
2   Department of Computer Science, Hekma School of Engineering, Computing and Informatics, Dar Al-Hekma University, Jeddah 22246-4872, Saudi Arabia
3   Department of Computing, University of Turku, FI-20014 Turku, Finland
4   Department of Technology, Higher Institute of Computer Sciences and Mathematics, University of Monastir, Monastir 5000, Tunisia
*   Correspondence: imed.bendhaou@utu.fi

**Abstract:** The Internet of Things is gaining more importance in the present era of Internet technology. It is considered as one of the most important technologies of everyday life. Moreover, IoT systems are ceaselessly growing with more and more devices. They are scalable, dynamic, and distributed, hence the origin of the crucial security requirements in IoT. One of the most challenging issues that the IoT community must handle recently is how to ensure an access control approach that manages the security requirements of such a system. Traditional access control technologies are not suitable for a large-scale and distributed network structure. Most of them are based on a centralized approach, where the use of a trusted third party (TTP) is obligatory. Furthermore, the emergence of blockchain technology has allowed researchers to come up with a solution for these security issues. This technology is highly used to record access control data. Additionally, it has great potential for managing access control requests. This paper proposed a blockchain-based access control taxonomy according to the access control nature: partially decentralized and fully decentralized. Furthermore, it presents an overview of blockchain-based access control solutions proposed in different IoT applications. Finally, the article analyzes the proposed works according to certain criteria that the authors deem important.

**Keywords:** Internet of Things (IoT); blockchain; access control; IoT applications; fully decentralized; partially decentralized

## 1. Introduction

Internet of Things (IoT) is a disruptive technology that resulted from the progress in sensor, communication, and embedded systems. It is a global network spawned from the evolution of the wireless sensor network [1]. IoT is a fully decentralized and heterogeneous system that has raised serious concerns about security and privacy. IoT architecture is composed of a perception layer responsible for the real-time collection of data from the environment. It is primarily realized using one or more ubiquitous sensors. The network layer is responsible for connecting the perception layer to the global Internet. In the context of fog/edge computing, this layer can also process and store data before sending it to the cloud server. The third layer is the application layer that processes, stores, and interprets data for subsequent actions [2].

Fog computing is a distributed computing system that has been advocated to address the shortcomings of cloud computing. The system, as depicted in Figure 1, is composed of end devices, fog nodes, and cloud servers. In the lower layer, the end devices/nodes represent many tiny and cheap devices (ranging into the billions) responsible for sensing and actuating. Fog devices that are in the middle layer have computation and storage

resources larger than end nodes. They also connect end devices to the cloud servers. The number of end devices in this layer is a few thousand. Finally, the top layer represents the cloud servers with extensive computing and storage capabilities. They are used to run heavy data analytic algorithms.

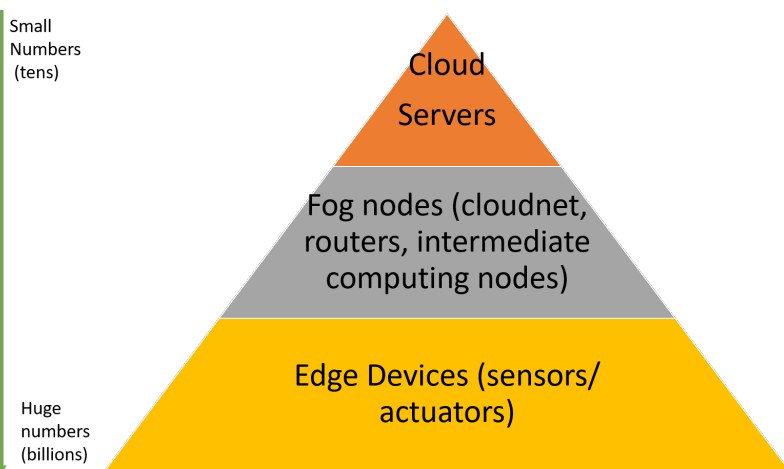

**Figure 1.** Fog computing layers.

To harmonize fog computing and to ensure the interoperability of hardware and software, IEEE has adopted the openfog reference architecture [3]. This latter is articulated around four pillars: data analytics, IT business and cross fog applications and control, performance, security, and manageability.

Parallel to the fog computing paradigm, dew and multi-access edge computing (MEC) were advocated to address the shortcomings of cloud computing. The dew-cloud architecture comprises a perception layer (sensors and actuators), a dew-server, and a cloud server [4]. The MEC is a computing paradigm proposed by the European Telecommunications Standards Institute (ETSI) that aims to bring cloud services near to the edge of the cellular network for context-aware applications [5].

IoT security has been identified by many scholars as a major concern for its effective widespread use [6] that has led IoT security shield to receive tremendous attention from the Defense Advanced Research Projects Agency (DARPA). In addition, ten research themes in IoT were identified by [7]. These themes have been clustered in three groups: security for end nodes; adaptive and context-aware security; and cognitive IoT security. As a reaction, some recent papers investigated the potential of machine learning (ML), artificial intelligence (AI), and blockchain to address the security issue in IoT [8].

Several attacks in wireless sensor networks (WSNs) were classified according to the following categories: selective forwarding attack, sinkhole attacks, wormhole attacks, Sybil attacks and man-in-the-middle attacks [9]. Furthermore, many types of attacks on IEEE 802.15.4 standards were attributed to three main groups, namely stenography attacks, message manipulation attacks, and radio jamming attacks. In [10], attacks were first classified according to the security goals that they wish to hit such as availability, integrity, and confidentiality. Then, the threats of information security were considered in the attack classification, namely: network attacks, host attacks, and application attacks. The most relevant attacks in IoT systems are listed in [11], namely: end device attacks, communication channel attacks, network protocol attacks, sensory data attacks, Dos attacks, and software attacks. It is obvious that attacks in the IoT domain can lead to disastrous consequences, for instance, hackers in one country can remotely gain unauthorized access to traffic light controllers and alter the traffic flow for their benefit [12]. In 2017, a ransomware attack on the colonial pipeline system in the US caused a huge economic loss and disrupted the fuel supply [13]. The ransomware attack is a very profitable cyberattack as victims have to pay the ransom money in exchange for the release of hijacked ICT resources or personnel

data [14]. Ransomware is a new generation of ransom attacks targeting wearable devices. Whatever the type of attack, it invades security services such as authenticity, integrity, confidentiality, and availability. These services can be protected using security mechanisms such as access control. This latter encompasses two phases, namely: the authentication and the authorization.

The authorization phase represents the process that specifies who can access particular resources and under which conditions. This represents an effective solution to prevent illegal access to IoT resources, such as smart devices and data. Traditional access control approaches are unable to give an effective mechanism to encounter the requirements of IoT systems. Additionally, most of these approaches use a centralized authorization server, which may generate an important communication overhead and involve high latency. In addition, using a single centralized authorization server that treats all access control requests can lead to a single point of failure (SPOF). To address these issues, many researchers used blockchain technology. This represents a P2P system that manages a distributed ledger. This latter can be used to store agreements, transactions, events, and smart contracts. The emergence of blockchain technology permitted the users to benefit from its properties, such as immutability, decentralization, anonymity, and confidentiality. Several surveys on the use of blockchain technology in access control were proposed. These papers did not present a deep analysis of the proposed solutions. Furthermore, none of these surveys mentioned in detail the phase of access control taken into consideration. Some of them presented the use of blockchain technology in several IoT applications but not, especially in access control. Additionally, most of these works neglected the existence of three categories of comparison criteria: some relating to blockchain technology, others relating to access control itself, and the last category that is concerned with the implementation and evaluation metrics. All the points mentioned above are motivating factors for the presentation of this work.

The main contributions of this paper are the following:

- A provision of a deep analysis of existing surveys on access control solutions that used blockchain technology to address the trusted third party (TTP) issue in an IoT environment.
- A background on access control and blockchain technology is presented to explain the importance of their combination to eliminate the use of a trusted third party (TTP) in an access control solution.
- A classification of the existing blockchain-based access control solutions according to their nature into two categories, namely: fully decentralized and partially decentralized.
- Recent blockchain-based access control frameworks are also classified according to the IoT applications. The analysis of these works according to the domain of application makes it possible to specify the outcomes of each domain.
- Blockchain-based authorization solutions are also analyzed according to certain criteria that we judge important.
- Open challenges that need to be addressed when designing blockchain-based access control solutions for IoT applications are also discussed.

This paper is structured as follows. Section 2 presents the recent surveys on blockchain-based access control solutions in IoT environments. Section 3 gives a brief background on the access control mechanism. Section 4 presents the blockchain technology and how it can be used in access control. Several works that deployed the blockchain concept in access control were also discussed in Section 4. Section 5 presents the most recent blockchain-based access control frameworks proposed in different IoT applications. Blockchain-based authorization solutions are analyzed in Section 6. Discussion and open issues are presented in Section 7. Finally, Section 8 concludes the paper.

## 2. Related Works

Numerous reports have discussed access control techniques. In this section, surveys pertaining to IoT, cloud computing, and wireless sensor networks (WSNs) are considered. Table 1 gives a brief description of the contribution and limitations of these survey papers.

**Table 1.** Related works' contributions and limitations.

| References | Year | Contribution | Limitations |
|---|---|---|---|
| [15] | 2014 | Authors presented a taxonomy of access control schemes in WSNs. | Authors only took into account the access control model as access control criterion. |
| [16] | 2019 | Authors classify access control solutions in a cloud computing environment using several categories. Additionally, they presented a performance comparison between different access control models. | Authors only managed the access control model as a criterion. The access control phases and nature were not handled. |
| [17] | 2019 | Authors presented several access control categories: one of them is related to the blockchain technology, namely: smart contract and transaction. Moreover, they specified the application domain and the used blockchain platform for each work. | Authors did not take into consideration the access control criteria such as access control models, phases, and natures. Furthermore, the implementation criteria encompass the blockchain platform, the hardware, and the performance criteria. |
| [18] | 2019 | Authors proposed a blockchain-based access control taxonomy: transaction-based access control and smart-contract based access control. | The survey neglected the comparison criteria related to access control and general criteria such as implementation and evaluation. |
| [19] | 2021 | Authors classified the existing related works according to several categories. | Authors did not make the difference between the criteria concerning blockchain technology and those relating to access control. |
| [20] | 2021 | Authors presented a blockchain-based access control taxonomy: transaction-based access control and smart-contract-based access control. | They only used two comparison criteria: implementation and security levels. |
| [21] | 2022 | Authors identified access control features for blockchain-based access control in IoT. | They only focused on some access control criteria such as attribute management and permission enforcement. |

Maw et al. [15] proposed a taxonomy for the classification of access control models used in WSN. The taxonomy clustered the schemes into three classes: role-based, cryptography-based, and privacy-preserving-based access control techniques. They further elaborated on two metrics to compare the access control schemes: the first metric is based on

the feature of the scheme (support for data/user privacy, flexibility, support for emergency data access, granularity, and context sensitivity), whereas the second one is based on the implementation performance. This includes computational overhead, energy consumption, and memory requirement.

Cloud computing is a paradigm shift in ICT (information and communication technology). It is a model in which computing resources (storage, network, services, servers, and applications) are shared among geographically distributed users or tenants. There are four categories of cloud computing: private, community, public, and hybrid. The positive and negative aspects of each model are discussed in Stalling and Brown, [22]. Cloud computing brought new challenges to legacy access control techniques [23]. The survey work of Cai et al. [16] discussed the subsequent models for access control: task-based, action-based, attribute-based, usage-based, and encryption-based access control methods. They further compared those models based on ten metrics: security, confidentiality, the flexibility of authorization, minimum privilege, separation of duties, fine-grained control, cloud environment attributes, constraints description, compatibility, and expansibility. At length, the authors identified the security of the virtual server, data set, and cloud platform.

The Internet of Things, IoT, is an emerging technology that connects objects, sensors, humans, machines, and living things using an all-IP network [24]. IoT is a disruptive technology that ignited the fourth industrial revolution, commonly known as Industry 4.0 or IIOT (Industry IoT). IoT has also been used in healthcare, transportation, agriculture, smart-city, retail, etc. Security of the IoT has received considerable attention from multiple stakeholders [25]. The legacy access control techniques (RBAC, CapBAC, and ABAC) are unfit for IoT [24]. To remedy this issue, blockchain-based access control has been proposed as a suitable candidate for IoT. In [17], Rouhani and Deters analyzed the issues of current access control solutions and explained how blockchain technology can handle these problems. Moreover, they classified the existing proposed solutions according to the following categories: transactions and smart contract category; data sharing category; cloud federation category; multiple organization category; blockchain category; and self-sovereign identities category. The authors also studied the application domain as well as the blockchain platform used in each solution.

In [18], Riabi et al. proposed a comprehensive review of the existing blockchain-based access control solutions. They classified these works into two categories, namely: transaction-based access control and smart-contract-based access control. They focused their comparison only on blockchain criteria: transaction or smart contract. Furthermore, the authors did not specify the access control phase for which the blockchain technology is used. Finally, Riabi et al. did not give information about the evaluation and implementation of their solutions.

A review paper on blockchain-based authorization in IoT was presented in Patil et al. [19]. Several categories were used in this article to classify recent works, namely the attribute-based access control (ABAC) category, the fair access category, the distributed access control category, distributed key management category, token-based access control category, control chain category, attribute update oriented access control category, ripple protocol consensus algorithm (RPCA)-based authorization category and multiple smart contracts-based authorization. In this paper, the authors give some other use cases of blockchain technology such as vehicular ad hoc networks (VANETs), healthcare, and supply chain applications.

In [20], Hussain et al. presented a survey on recent blockchain-based access control schemes. They used two groups to classify these solutions. The first group is concerned with access control based on blockchain transactions. The second one is relative to the solutions that used the smart contract technology. Additionally, the authors give the positive and negative aspects of each solution while defining its level of security. The implementation criterion was also taken into consideration in this survey.

The recent work of Shantanu et al. [21] summarized the advantages of blockchain-based access control and compared recent works using five features: permission enforce-

ment, access rights transfer, resource management, scalability, and attribute management. The authors reported that further work is needed in standardization, policy, identity, and trust management.

After a deep analysis of Table 1, it is clear that most of the existing survey papers presented a taxonomy of the blockchain-based access control solutions. This taxonomy is concerned with two different concepts, namely: blockchain technology and access control itself. Blockchain technology has its criteria that permit us to evaluate the effectiveness of the access control model. To illustrate, it is suitable to give the example of two blockchain-based access control solutions, one only using transactions for the creation of security policies and the management of access requests. The second one uses a smart contract with transactions. This last combination makes it possible to reduce the response time to an access request because it eliminates the search for security policies on the blockchain. On the other hand, access control has its criteria that can influence the effectiveness of the solution. For instance, the choice of access control model influences the efficiency of the solution. When taking the case of the RBAC model, its inadequateness for a distributed and large-scale environment such as the IoT is detectable. It is fair to say that a survey on blockchain-based access control solutions must consider all the necessary criteria relative to both blockchain technology and the access control process. This assumption will be highlighted and discussed in this article.

## 3. Access Control Techniques

In this section, a discussion of the access control techniques is presented from two point of views. In the first subsection, there is an overview on the access control evolution. Followed by a definition of some important concepts. In the second subsection, the access control is also defined and associated with more details on its two phases, namely: authentication and authorization.

### 3.1. Legacy Access Control Methods

Access control is an important security mechanism that emerged during the past century. The most obvious application of access control is the protection of properties, people, and areas. Physical access control systems (PACSs) have undergone major transitions from purely mechanical to more sophisticated systems (biometric systems). Figure 2 illustrates the evolution of the physical access control method from metal keys to biometric systems (fingerprint and face recognition). Traditionally, access control to private space is granted by using metal keys. The mechanical system was later replaced with an electronic lock which enabled the use of numerous authentication mechanisms: keypads, proximity cards, and biometrics (e.g., face recognition and fingerprint). Access to IT resources can be controlled using PACS.

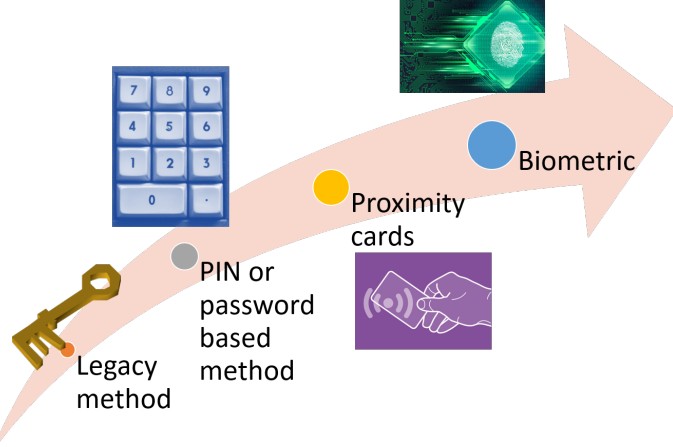

**Figure 2.** Evolution of access control systems from metal keys to biometric systems.

Access control methods have also been applied to secure computer systems. Before going deeper into this subject, here is a restatement of the definitions described in the National Institute of Standards and Technology (NIST) and Stalling and Brown [22].

**Definition 1.** *A subject is an entity (human, organization, system, process, an app, etc.) that can interact with an object.*

**Definition 2.** *An object is an asset (sensor, CPU, memory, files, application, subject, etc.) with restricted access.*

The access rights for each subject are recorded in an access control matrix. An example of a matrix is shown in Figure 3. The example shows three subjects (S1, S2, and S3) with six objects. The subjects are also called an object. The entry in each matrix describes the access rights for each subject to each object. For example, subject S1 can read the object file, use the printer, and put the process in sleep mode.

|  | s1 | s2 | s3 | File | Process | Printer |
|---|---|---|---|---|---|---|
| S1 | control |  | Owner | Read* | Sleep* | Print |
| S2 |  | control |  | Execute | Wakeup |  |
| S3 |  |  | Control | Read/ Write |  | Print |

∗ : the right can be transferred to another subject

**Figure 3.** Example of an access control matrix.

The presence of a flag (star in Figure 3) in the access right means that the subject can transfer this right to another subject. This operation triggers the update of the access control matrix. For example, subject S1 can assign Read or Read* to the subjects S2 and S3. The subject who owns an object can grant access to other subjects. A detailed explanation of the access control matrix is in Stalling and Brown, [22].

The legacy access control policies are attribute-based access control (ABAC), role-based access control (RBAC), mandatory access control (MAC), and discretionary access control (DAC).

*3.2. The Access Control in IoT*

Access control techniques require two essential phases, namely authentication and authorization. Authentication is the step that checks whether an individual is presenting the correct identity to detect any attempt of the impersonation of users. Authentication is a good first step, however, it is not sufficient because an authenticated user can still try to perform actions that are not allowed. Additionally, the authorization step guarantees that an authenticated user can only perform the authorized actions. It can be of three different types, namely: policy-based, token-based, or cryptography-based. These types can be combined to have: policy and cryptography-based or cryptography and token-based approaches.

A policy-based authorization requires enduring several phases. The first one is known as a security policy definition. It permits the specification of a set of rules that describe the security requirements of the resource owner (data or device). Then, an authorization model is used to encapsulate the defined policy. There are several models in the literature such as the RBAC model [26] where the rules are defined using the role of the user in the system. There are other authorization models including the mandatory access control (MAC) model, the dictionary access control (DAC), and the capability-based access control (CapBAC) model. However, all the models mentioned above do not make it possible to guarantee the security requirements of a distributed and heterogeneous environment

such as the Internet of Things (IoT). This type of environment needs an access control system that addresses many requirements through which it can be mentioned: scalability, interoperability, availability, and dynamicity. When taking the case of the classical RBAC model into consideration, it is possible to claim that specifying security policies for complex IoT scenarios is inappropriate. In [27], Sun et al. employed the user's location context to extend the RBAC model. On the other hand, the attribute-based access control (ABAC) [28] is a model that permits the specification of the security rules based on several attributes. The latter represent the pertinent features of the environment, resources, and subjects. The ABAC model is more suitable for a large-scale distributed environment such as IoT. Furthermore, it permits guaranteeing a real-time access control process.

The ending step of the policy-based authorization consists of choosing the suitable architecture to implement the access control model using the appropriate mechanisms that specified in what way access requests are evaluated against security policies, such as cryptography and access control lists (ACL). A token-based authorization has two essential steps: token creation and token validation. Eventually, cryptographic authorization gives selective access to encrypted data using a combination of key management and cryptographic schemes, namely attribute-based encryption (ABE).

## 4. Blockchain-Based Access Control

The main aim of this paper is to show how blockchain technology can be used in IoT access control solutions. Consequently, a detailed description of blockchain technology is presented in the first subsection, whereas the second one discusses the IoT and blockchain integration. Moreover, the use of blockchain technology in access control is presented in the last subsection. In the latter, a taxonomy is presented dividing related works into two classes according to the authorization nature, namely: fully decentralized and partially decentralized.

### 4.1. Overview of the Blockchain

The blockchain is a secured, distributed, and shared ledger that can keep track and record any type of transaction without the need for any centralized entity [29]. Bitcoin represents the first generation of blockchain. It was initially introduced as a cryptocurrency by Nakamoto in 2008 [30]. Then, the second generation of blockchain appeared with a new concept called a smart contract. This latter represents programs stored on the blockchain that are operated when predetermined conditions are met. The following generation evolved with decentralized storage and decentralized communication such as Ethereum swarm [31]. Finally, the last generation of blockchain presented solutions to introduce this technology in industry and business.

Blockchain can be divided into three categories according to the restrictions defining how users can interact with the ledger. First, a public blockchain is also known as a permissionless blockchain where everybody can read, write, and verify the blocks. Additionally, there is the private blockchain, which is available to a certain group and limits access rights. The final category is the consortium blockchain where more than one person is managing the network. This category represents an extension of the private blockchain.

Regarding the structure of the blockchain, it encompasses three essential portions, namely: block, chain, and network. A block contains a header, hash value of the previous and current block, timestamp, and transactions (illustrated in Figure 4). The chain is used to link blocks such that each block can access all other blocks [31].

The blockchain can also be considered as a peer-to-peer network because it does not need a central entity that decides to store and manage data. For the fact that these tasks are handled by peers doing the mining, this latter is the process that permits the addition of the validated transactions in blocks (Figure 5 summarizes the steps undertaken to add a new block in the ledger). Additionally, miners are responsible for broadcasting this block to all nodes of the network.

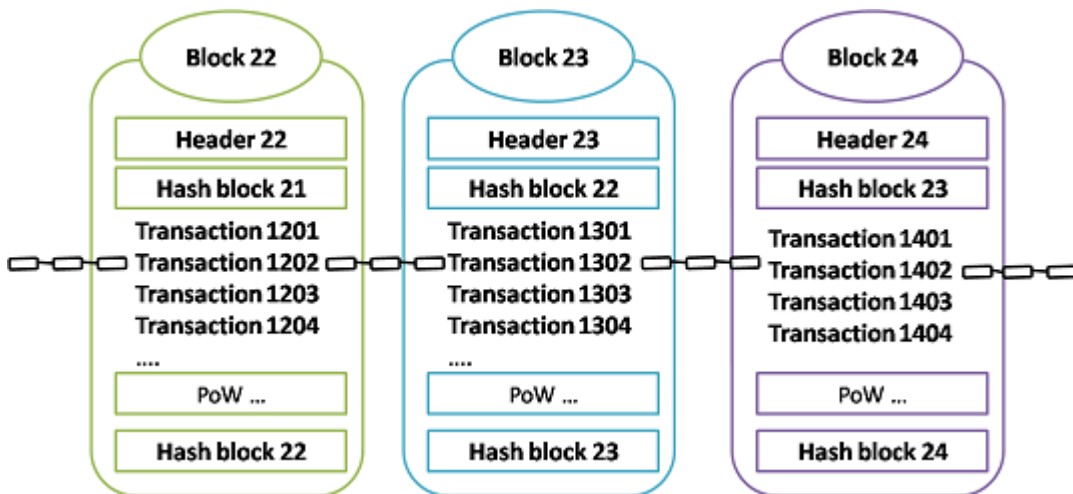

**Figure 4.** A blockchain structure.

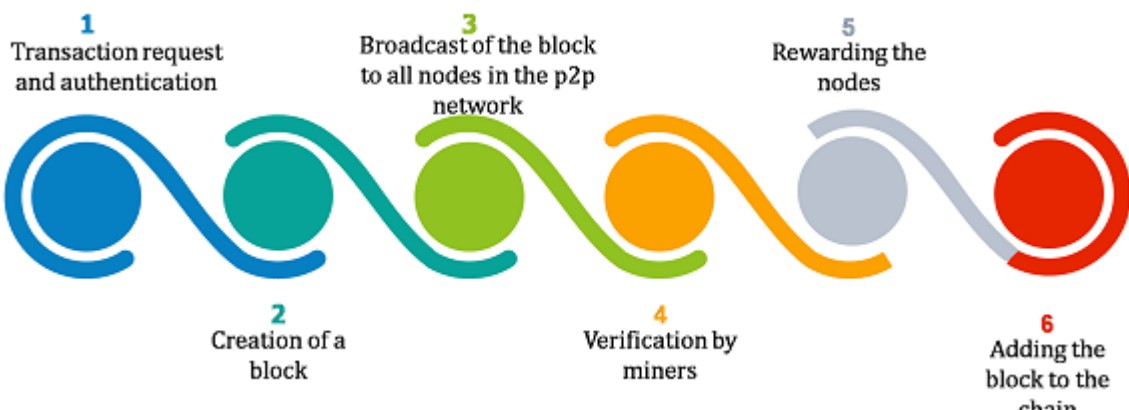

**Figure 5.** A blockchain process.

The mining process is executed through a consensus algorithm. The proof of work (PoW) is largely used by several blockchains to validate transactions. In this algorithm, a mathematical puzzle is solved by miners using a considerable amount of computational power. The first miner who solves the problem will receive a reward for their work. To overcome the greater energy consumption of PoW, a proof of stake (PoS) algorithm is proposed. This algorithm favors nodes that have a high stake. These nodes can add blocks to the blockchain. To reduce the consumption in mining, a random selection using the amount of stake is accomplished to select a new block. A variable of the PoS algorithm is presented as the delegated proof of stake (DPoS). It selects delegates that give blocks according to an attributed order. The DPoS algorithm reduced the computational cost because the number of blocks is lower than other algorithm's number of blocks. Additionally, the delegates are disqualified if they give invalid blocks or lose them. A practical Byzantine fault tolerance (PBFT) was presented. The principal aim of this algorithm was to address issues of Byzantine fault tolerance. This takes into consideration the existence of malicious nodes in the system. In this algorithm, nodes are divided into two categories: a primary node known as a leader, and secondary nodes. PBFT requires that the number of harmful nodes must not exceed one-third of the nodes' total number in the system. Although this algorithm does not waste energy, it is weak versus Sybil and scaling attacks.

### 4.2. Blockchain and IoT Integration

Today's IoT systems consist of a large number of interacting devices that can exchange collected data and remotely control objects across the Internet. These devices are known

for their reduced storage and computing capacities. Furthermore, they use a central server that handles authentication and authorization. It should be noted that heterogeneity and decentralization are two important features of the IoT environment due to the interaction of end devices and networking technologies [11]. Consequently, the data generated by IoT devices need to be protected from unauthorized users. Alternatively, blockchain as a decentralized technology can eliminate the concept of a central entity improving scalability and transparency. Additionally, the cryptographic structure of the blockchain permits ensuring data privacy and confidentiality. Indeed, the use of the smart contract with blockchain technology eliminates human interference [11]. From the above, it is obvious that the integration of blockchain technology with IoT can improve IoT issues. Thus, Saxena et al. [11] enumerated several benefits of this integration, namely enhanced security, improved interoperability, autonomous interactions, reliability, and secure code deployment. Additionally, the challenges encountered when integrating the blockchain with the IoT were discussed in [31]. Blockchain has several applications in IoT comprising banks and insurance, healthcare, commerce, food industry [32], smart grid, and security [33]. The security improvements in IoT using blockchain technology is discussed in [11]. Authors confirmed that the enforcement of authorization policies in IoT using blockchain technology enhances the overall network security. The following subsection is dedicated to blockchain-based access control.

### 4.3. Access Control Using Blockchain

As explained previously, access control techniques go through two different steps, namely authentication and authorization. Some recent works proposed solutions that used blockchain technology in both steps, whilst others have not (illustrated in Figure 6). In this article, the focus was on blockchain-based authorization to reach the aspiration of showing the advantages of this technology in the security of an environment such as IoT. The authorization step can be parted into three types: policy-based, token-based, or cryptography-based. It can also combine these previous types or combine one of them with other methods such as permission delegation.

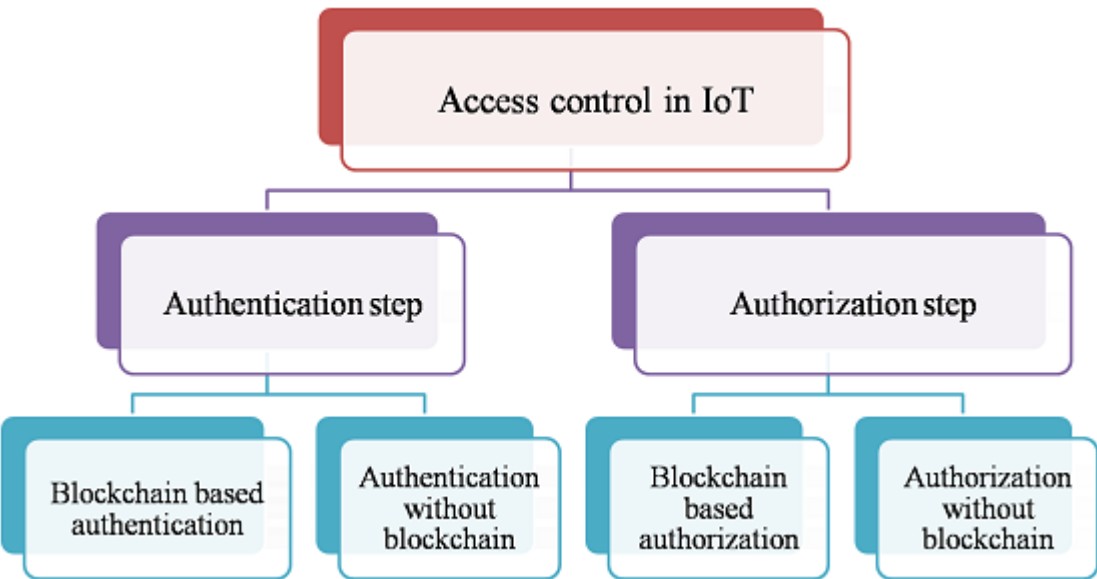

**Figure 6.** Access control steps.

A policy-based authorization consists of two essential phases: the specification of security policies and the evaluation of these policies upon receipt of an access control request. Correspondingly, a token-based authorization needs two fundamental procedures, where the first one permits the creation of the token, whilst the second step is concerned

with the token's verification. Furthermore, cryptographic authorization permits selective access to encrypted data via a combination of key management and cryptographic schemes.

The nature of an access control model depends on the number of entities that participate in each step of the process. In traditional access control, the system administrator inserts the security policies into the policy database. In addition, one decision point manages the evaluation of an authorization request. It is fair to say that this type of process is considered a centralized access control that does not meet the security requirements of a heterogeneous and distributed environment, such as the Internet of Things (IoT). To manage the issue cited above, recent works about access control have focused on a decentralized approach to ensure scalability while guaranteeing a certain level of privacy. Fortunately, the emergence of blockchain technology, with its decentralized nature, has greatly contributed to ensuring this type of authorization. Hence, according to the previous definition, it is possible to claim that an access control process using blockchain technology is even partially decentralized or fully decentralized.

In the case where several entities participate in the specification of security policies and their evaluation, the blockchain-based authorization can be classified as fully decentralized, otherwise, it will be considered partially decentralized. Regarding a token's based access control, it is considered as a partially decentralized process upon condition that one entity creates the token and another entity verifies it. Eventually, a distributed key-based process is considered as a fully decentralized. Figure 7 illustrates the categories and subcategories of authorization based on blockchain technology in an IoT environment. In this sub-section, a classification is provided regarding the different related works based on two categories: fully decentralized and partially decentralized.

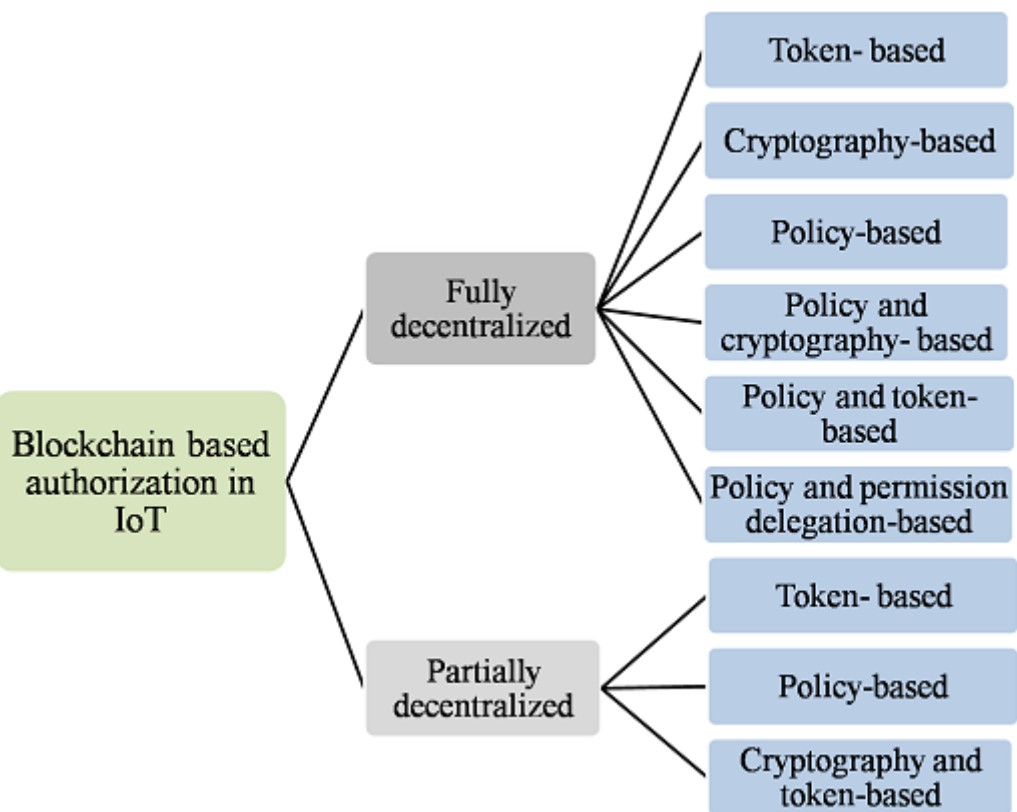

**Figure 7.** Taxonomy of authorization based on blockchain technology in IoT.

Partially Decentralized Authorization

In [27], Sun et al. presented a role-based access control (RBAC) model using a private blockchain. A multi-index table and key-value database were employed to create the access control policy and stock the summary of access control information on the blockchain. In

addition, a smart contract manages the user's access queries. At last, authors chose the distributed proof of stake (DPOS) as a consensus algorithm in the blockchain. A private blockchain was combined with an MAC policy.

In [34], Novo addresses the scalability problem of access control in an IoT environment using a private blockchain to store and manage access control information. The proposed solution is partially decentralized for the fact that a manager defines the access control rules and administers the permissions through a single smart contract. A special type of nodes named management hubs were used to link multiple systems to the blockchain. The proposed scheme offers the possibility of updating and deleting authorization policies. There is an absence of support concerning the safe insert of devices.

In [35], Ourad et al. presented a solution that used the blockchain technology to ensure authentication and secure communication in IoT devices. An Ethereum blockchain was used with a smart contract deployed by an admin user. For further elaboration, a login-admin is a method of the smart contract that permits creating the user's tokens. The principal use of smart contract in this work was to reduce the authentication overhead. For this purpose, the authentication phase is executed through the smart contract. In the case where the user is authenticated, the smart contract will also decide whether the user is authorized to access resources. In this previous case, the smart contract broadcasts an access token to the user and to the IoT device. Hence, the token verification is executed through several IoT device; however, the token creation is executed through the smart contract.

In [36], Algarni et al. used a multi-agent architecture to guarantee secure access control in IoT system. The main goal of the solution was to use different blockchain managers (BCMs) that manage the IoT entire system. Furthermore, these managers set the access control policy for each device in the different layers of the system. This created the possibility to classify the solution as a partially decentralized access control. Although the proposed system allows securing communication between the fog nodes, cloud computing and IoT devices, it used a big header size in the blockchain and it has not yet been implemented.

Hierarchical blockchain architecture for access control in IoT environment was presented in Abdi et al. [37]. The proposed solution reduced network overhead and transaction latency where authors used three types of managers. An edge blockchain manager (EBCM) validates and grants access to devices that need to reach or store data in the same cluster. Furthermore, Abdi et al. introduced an aggregated edge blockchain manager (AEBCM) to authorize requests from sub-device clusters and manage the security policy of the system, which is based on the attribute-based access control (ABAC) model. At this level, the blockchain uses a practical Byzantine fault tolerance (PBFT) as a consensus algorithm. By the end, a cloud consortium blockchain manager (CCBCM) authenticates users and selects the corresponding ABAC policy; after evaluating the request, the manager returns the response to the user. Several smart contracts were used in different layers of the system to store the access control rules and information of IoT devices.

Islam et al. [38] proposed a federated learning-based data accumulation solution. This latter combines drones and blockchain technology. Additionally, it handles the authentication and authorization phases. To guarantee two-phase authentication, a cuckoo filter and a timestamp nonce were used. Moreover, the scheme can be classified as a token and cryptography-based technique. It is partially decentralized for the reason that the token is created by the edge server closest to the entity.

### 4.4. Fully Decentralized Authorization

Many researchers were able to guarantee totally decentralized access control. In [39], Xu et al. combined the policy-based access control with a token-based approach. They used a private blockchain with smart contract technology to ensure an effective access control scheme known as BlendCAC. The proposed solution is fully decentralized because the domain owner defines the security policies of its resources. Additionally, the access request

evaluation (policy evaluation and token validation) is executed among a large number of devices.

In [40], Zhang et al. used multiple access contracts to achieve a fully decentralized policy-based access control in an IoT environment. They combined machine learning algorithms with smart contracts to check the behavior of the subject. Each access control contract defines access control policies for the subject–resource pair. Each object has the ability to create a new access control method. Additionally, the proposed solution took into consideration the case of the modification of an already existing policy. Finally, Zhang et al. implemented and validated the proposed framework.

In [41], Islam and Madria considered fully decentralized access control, where all the access control phases (creation of policies as well as their evaluation) are carried out based on a consensus of several entities. They designed and implemented the attribute-based access control (ABAC) model in a private blockchain named Hyperledger Fabric. A smart contract was deployed to manage the security policy, which the requester and the owner have agreed on.

In [42], Zhang et al. used a consortium blockchain to achieve a collaborative attribute-based access control model for IoT devices. The proposed solution includes five components that are: authority nodes (AN), consortium blockchain network, IoT devices, public ledger, chain-code, and access tree. In the access control system, each device can define its own security policy. The latter will be translated to an access tree. Furthermore, authority nodes are built to make access control decisions and interact with the blockchain. Finally, the evaluation of the proposed scheme has shown that it is efficient for it reduced the computation and storage overhead.

In [43], Nakamura et al. focused on the critical access control issue for IoT devices using Ethereum smart contracts to store and manage capability tokens. In the proposed CapBAC scheme, the owner entity creates a capability token. Apart from that, the owner entity can delegate tokens to other objects. Ultimately, the last step of the authorization process represents the token verification. In the proposed solution, the subject is responsible for passing the token to the smart contract to verify whether the access is accepted or denied. Hence, the access control system is fully decentralized because the token creation and delegation are executed by several entities. Furthermore, several subjects execute the token verification. A fully decentralized key distribution access control system in IoT-enabled smart grid environment was presented.

In [44], Bera et al. used a private blockchain to secure data transfer between smart meters and service providers. The proposed solution used the practical Byzantine fault tolerance (PBFT) algorithm as a consensus mechanism with a secure leader selection to add blocks to the blockchain. Furthermore, the Elliptic curve cryptography (ECC) [45] algorithm was used to encrypt transactions. Eventually, the authors demonstrated that their solution was secure through several types of verification.

In [46], Zhai et al. used the attribute-based access control (ABAC) model with the smart contract to achieve a fully decentralized policy-based access control system applicable in the process of intelligent manufacturing. They used formal language to represent access control attributes. Furthermore, in the proposed solution, security policies are self-created by each resource owner. A requester obtains access to a resource provided that all its attribute–value pairs and the current environment's attributes meet the access policy requirements set by the resource owner. The latter executes the evaluation of an access control request for each smart contract it invokes..

To overcome the inflexibility and the key abuse problems of the attribute-based encryption (ABE) scheme, Gu et al. proposed in [47] the TABE-DAC scheme, which can be classified as a policy and cryptography-based solution. It has the feature of being fully decentralized due to the fact that data owners create and update their security policies. Additionally, these policies are verified by several entities. Hence, the proposed solution is based on cryptography and blockchain technology, which offers the possibility of tracing

malicious users. Finally, the authors evaluated the effectiveness of the proposed scheme theoretically.

In [48], Ali et al. presented a blockchain-based permission delegation authorization model. In the proposed solution, they took into consideration two types of access control requests, namely the event and the query ones. The first one is concerned with the request generated in a response to an event. The second type, however, deals with the access control request generated by a user to access a certain resource. In addition, the authors managed the permission delegation for the two types using a private blockchain with smart contract technology.

## 5. Applications

The Internet of Things paradigm has large applicability in several areas, such as smart farming, smart healthcare, smart grid, smart transportation systems, and industry 4.0. All these fields of application have one thing in common, which is the sensitivity of the data generated by smart devices. It is clear from previous sections that traditional access control techniques are inadequate for this type of environment. In addition, blockchain-based access control makes it possible to guarantee a partially decentralized or fully decentralized access control that meets the security requirements of an IoT environment. This section will present the different blockchain-based access control techniques that have been proposed for each area. Then, an analysis of the limitation of these works concerning the requirements of the area will be presented. In the end, possible solutions for these limitations are provided.

### 5.1. Smart Farming

The emergence of smart farming has enabled farmers to manage their farms in a better way. Compared to traditional farms, smart farming requires communication between different intelligent objects to achieve a good farming result. Although the use of these connected objects has promoted the process, on the other hand, it has attracted the attention of some malicious users. The latter can generate cyber-attacks to remotely control smart objects, such as reapers, tractors, soil moisture sensors, temperature sensors, trucks, and harvesters. An access control process remains the means that makes it possible to avoid inadequate access. This subsection will present the different blockchain-based access control techniques in a smart farming environment.

In [49], Arshad et al. presented a private blockchain-based access control (PBAC) scheme in a smart agriculture environment. They used a private blockchain that is managed and maintained by an administrator. Furthermore, the main approach of this solution is the use of an access control list with a policy header. The access control list saves the identities of legitimate visitors and access levels that the owner permits. Additionally, the authors store in the blockchain transactions of access requests, which targeted different devices. Finally, the evaluation of the proposed scheme is realized and showed that the protocol is secure against several malicious attacks.

In [50], Vangala et al. focused on the authentication issue in the smart farming environment. They used smart contract and blockchain technology with edge computing to ensure authentication at two different levels, namely device-to-gateway authentication and device-to-device authentication. Finally, the evaluation of the proposed authentication scheme showed that it is effective against several attacks compared to other schemes.

### 5.2. E-Health

The most important objective of a smart health environment is to allow medical practitioners to guarantee better healthcare services. In a smart health system, smart devices collect important data on patients' health. These data must be transferred and stored securely. Additionally, they must be only accessed by legitimate users, because allowing access to unauthorized users can have serious consequences on the health of the patient.

In [51], Sookhak et al. focused on EHR access control methods. They presented the most recent works that used smart contracts with blockchain technology in the healthcare domain. Moreover, several parameters such as consensus protocol, type of EHR storage, type of blockchain, and type of ledger were used to present a blockchain-based access control taxonomy. Finally, after analyzing the proposed blockchain-based access control methods in the healthcare domain, the authors found that using a smart contract by the data owner to define security policies is costly in such an environment because the owner has to update the smart contract to guarantee user revocation. Furthermore, ensuring this feature causes a high computation overhead on the blockchain. Sookhak et al. noticed that several works were presented to overcome this issue. On the other hand, some of the proposed reports have tried to resolve the data storage issue by protecting the EHRs before outsourcing them into the cloud. The authors mentioned several works on blockchain scalability wherein it was managed at different layers, such as a scalable IoT architecture or a scalable consensus protocol (PoW, BFT). Finally, the last-mentioned issue cited in this paper is latency in the blockchain network, which has a significant impact on the patient's life.

In [52], Hossein et al. proposed the BCHealth architecture that allows data owners to specify their security policies over their data. For that purpose, the authors used two chains, where the first chain was used to store data transactions and the second one concerned access control policies' storage. To reduce the communication overhead, the authors chose to store healthcare data on a machine near the data owner. This point represents one of the advantages of this solution because most of the previous works stocked data in the cloud or in the healthcare data center. Finally, the authors used a proof of authority algorithm (PoA) as a consensus algorithm. Although the use of two blockchains has improved network performance (reduction in transaction search time), the authors gathered the blockchain network nodes into several clusters and attributed each user to a specific cluster for stocking their access control policies and data. Consequently, cluster management remains an issue to be resolved.

In [53], Zhang et al. presented a blockchain-based hierarchical data-sharing framework (BHDSF) to give a fine-grained access control in the healthcare Internet of Things (H-IoT). The authors implemented Ciphertext-policy attribute-based keyword search (CP-ABKS) to reduce the burden of traditional CP-ABE schemes for PHR searching. Additionally, they deployed a user hierarchy to delegate key distribution. One of the limitations of the proposed solution is the data-sharing delay, which can be reduced using edge computing.

In [54], Yang et al. proposed an efficient attribute-based encryption (CP-ABE) scheme in the access control of patients' electronic health records (EHRs). They used edge computing to improve health services and reduce transmission delay. This delay was decreased by transferring encryption and decryption parts to the fog nodes. Moreover, the authors used the blockchain technology with a smart contract to guarantee a the fine-grained access control of the data. One of the limitations of this solution is the use of a key generation center that represents a single point of failure and can cause key escrow problems.

*5.3. Intelligent Transportation System*

Smart transportation system represents a major branch of smart cities. It can resolve several problems, such as safety [55], car parking, and traffic congestion. Traffic lights are a simple example that permits understanding how smart devices can make transit safer and faster. On the other hand, allowing dishonest people to access these connected objects can lead to serious consequences. Let us take the example wherein malicious persons are able to access traffic lights. They can provoke severe accidents. Hence, the importance of fast and effective access controls the process in this area of the IoT. In this section, different blockchain-based access control approaches that were proposed in the domain of smart transportation will be summarized.

In [56], Dukkipati and Zhang presented a blockchain-based access control technique. They used the attribute-based access control (ABAC) model with extensible access control markup (XACML) language to represent security policies. In this solution, the authors used two types of policies: general policy and special policy. This classification aims to reduce the burden of creating the same type of policies for each user. To permit security policy update and modification, Dukkipati and Zhang stored the policy link on the blockchain instead of storing the security policy itself. Finally, for the evaluation of the proposed approach, the authors considered the requisite access control policies for information sharing, regarding free parking slots, traffic signals, and traffic flow between two light signals.

In [57], Hu et al. proposed a blockchain-based parking management framework. The targeted issue of this solution is to preserve the privacy of users without the use of a trusted third party (TTP). They combined smart contract technology with the block chain open source (BCOS) to allow the sharing of parking spaces. The proposed system encompasses three layers, namely the storage layer, the management layer, and the user layer. The storage layer represents the blockchain wherein data are stored. The management layer is responsible for the authentication process. It is composed of authentication parties, which are considered the privileged users of the consortium blockchain. Finally, the user layer considers two types of users: vehicle drivers and the owners of parking lots. The two types of users can insert data into the system.

In [58], Amiri et al. presented a blockchain-based smart parking management system. In the proposed solution, several parking lots created a consortium blockchain to stock the available services and parking availability. First, every parking lot has to send its parking offers to the blockchain network. A distributed ledger where the blockchain network notes the offers is shared. Additionally, when the driver sends a transaction to the network to collect parking offers, the driver's location is masked using the cloaking technique. As such, the authors guarantee the drivers' privacy. Furthermore, after the selection of the best offer based on some criteria, drivers use Bitcoin to pay for parking services. Eventually, it is obvious that the presented scheme is based on authorized validators (parking lots), which manage the parking services.

*5.4. Smart Grid*

The smart grid is a promising technology that can give effective electricity to consumers in a secure way. Compared to traditional power grids, the smart grid permits having two-way information circulation between the power provider and the user. Smart grid networks face several security issues, such as unauthorized access to resources and the need for a third party that manages the authentication and authorization processes. Blockchain technology can be used to resolve these issues because it is based on a decentralized architecture with distributed computing. These latter features guarantee the security requirements of a smart grid environment with massive user access. In this sub-section, the most important solutions that were proposed to treat the issues mentioned above will be presented.

In [59], Zhou et al. presented a blockchain-based access control scheme adapted to several scenarios in the smart grid environment. The proposed solution used a combined cryptosystem. Moreover, it permits the users to participate in the smart grid's management. Additionally, to fix the key escrow problem of the untrusted third parties, the authors designed a consensus algorithm for the selection of a private key generator (PKG). Finally, the authors reduced the communication costs by reusing saved keys. Data generated by smart electrical devices are sensitive because they may contain the private information of consumers.

To resolve this issue, in [60], Le et al. proposed a fine-grained access control scheme based on Ciphertext-policy attribute-based encryption (CP-ABE) in the smart grid environment. The proposed solution used blockchain technology to manage logs with non-repudiation while ensuring their integrity. Compared with predecessor solutions, the

evaluation of the proposed approach showed that it ensures efficient communication with more security features and several functionalities.

In [61], Yang et al. affirmed that Ciphertext policy attribute-based encryption (CP-ABE) is suitable for the distributed environment. On the other hand, the limited computing capability of IoT devices prevents them from supporting the computationally intensive nature of ABE algorithms. To overcome this issue, the authors presented an edge blockchain-empowered secure data access control solution in the smart grid environment. For that purpose, the authors outsourced the end-users computation workloads to the edge nodes in a consortium blockchain system. Moreover, the authors used the threshold secret sharing scheme to avoid the single point of failure problem of the centralized authority.

In [62], Nasser et al. affirmed that smart meters represent the most vulnerable devices in a smart grid environment. That is why most recent research has focused on how to ensure secure communication between smart meters and electricity providers. In addition, they found that malicious activities and data alteration are the most important issues that smart grid environments must face. To address these latter issues, the authors presented a blockchain-based decentralized lightweight access control scheme for a smart grid environment. The proposed solution used the elliptic curve cryptography (ECC) [45] to guarantee secure communication between smart grid entities. Additionally, combining blockchain technology with a smart contract permits ensuring non-repudiation while improving grid reliability. On the other hand, the authors took into consideration the smart meters authentication. They proposed an identification protocol to manage this phase. Finally, the evaluation of the proposed scheme demonstrates that it is effective in terms of communication and computation costs.

*5.5. Industry 4.0*

Industry 4.0 is a new emerging paradigm that offers opportunities for better decision-making in the industrial field. The Industrial Internet of Things (IIoT) is also defined as a network with billions of industrial factories and machines equipped with sensors. These devices are connected to the Internet for collecting and sharing data. That is why it contributes to the increase in new risks and security challenges, such as malicious access, data tampering, and several cyber-attacks. This subsection will present the most recent works that used blockchain technology to ensure and improve security in Industry 4.0 environments.

In [63], Wan et al. integrated the blockchain technology in the Industrial Internet of Things (IIoT) architecture for smart factories. They proposed that the solution is based on five layers, namely the sensing layer, the management hub layer, the storage layer, the firmware layer, and the application layer. Each layer plays an important role in ensuring the best functioning of the system. Furthermore, the authors used the SHA-256 [64] and the elliptic curve cryptography (ECC) [45] to guarantee the privacy of the data generated by sensors. Regarding the access control process, the authors used an access control list that combines the Bell–La Padula (BLP) [65] model with the Biba model [66]. This combination permits maintaining the confidentiality, integrity, and availability (CIA) requirements.

In [67], Lahbib et al. proposed a privacy-preserving distributed access management framework (PDAMF). In the proposed solution, the authors used blockchain technology with ring signatures to ensure the access requester's anonymity. Additionally, they took into consideration both the authentication and the authorization phases. For this latter, they used a smart contract that evaluates the access request by checking the validity of the requester's role. On the other hand, the smart contract must verify the existence of the access control policy among the predefined security policies. The evaluation of the proposed scheme shows that it is suitable for an Industry 4.0 environment.

In [68], Feng et al. combined a consortium blockchain with the 5G technology to present a secure access control framework in the Industrial Internet of Things (IIoT) system. They used three types of chain codes, namely policy management chain code (PMC), access control chain code (ACC), and credit evaluation chain code (CEC). The PMC chain code is

used to specify the security policies of the objects. These policies are based on the attribute-based access control (ABAC) model. Only the object owner can execute the PMC. The ACC chain code plays the role of a policy decision point (PDP). First, it checks the existence of the policy that is needed to evaluate the access request. Then, if a security policy is found, the ACC evaluates the policy using the attributes' values and returns the access control response. Regarding the CEC, Feng et al. used credit evaluation criteria to dynamically select the order nodes. This last selection permits improving the security of the process. Finally, the evaluation of the proposed scheme showed that the use of the credit criterion improved the security of the access control process. Furthermore, the proposed framework reduced resource and communication consumption compared to the practical Byzantine fault tolerance (PBFT)-based scheme.

In [69], Shih et al. deployed three types of smart contracts with blockchain technology to present a distributed access control in the environment of the Industrial Internet of Things (IIoT), starting with The first type of smart contract, namely the policy contract (PC). The latter deals with the creation of a security policy using the attribute-based access control (ABAC) model, its validation, as well as its deletion. Furthermore, the authors used a device contract (DC) responsible for stocking the device URL in the ledger. This contract can also generate a one-time device URL to ensure the security of the shared data. Eventually, the authors used an access contract (AC) that evaluated the access control request and returned the permissions. The evaluation of the proposed solution showed a less requirement of time compared to proof of work (PoW) to reach a consensus.

## 6. Analysis of Blockchain-Based Authorization Frameworks for IoT

Numerous blockchain-based authorization solutions were presented in the literature. A review is provided in this subsection about existing solutions along with an analysis according to three groups of criteria defined in Table 2. The first group is related to criteria regarding blockchain technology. In particular, Cr1, Cr2, Cr3, and Cr4 are identified. While the second group encompasses the criteria that are related to access control, namely Cr5, Cr6, and Cr7. At last, the third group contains two general criteria that are: Cr8 and Cr9.

**Table 2.** Evaluation criteria.

| Category | Criterion ID | Criterion |
|---|---|---|
| Blockchain | CR1 | Number and type of blockchain |
| | CR2 | Type of consensus algorithm |
| | CR3 | Smart contract use and number of contracts |
| | CR4 | Blockchain platform |
| Access control | CR5 | Access control nature |
| | CR6 | Access control phases |
| | CR7 | Access control models |
| General | CR8 | Domain application |
| | CR9 | Implementation and evaluation criteria |

1.　　Cr1: number and type of the blockchain

After a deep analysis of the blockchains used in the proposed access control solutions (summarized in Table 3, it can be stated that no solution used a public blockchain. Some authors deployed a private one in [27,34,36,41,44,49,52]. In parallel, others used a consortium blockchain in [37,42,57,58,61,68]. This choice of type is determined by the nature of the information relating to the access control process. This information is generally confidential and sensitive, with an obligation of not being publicly accessible. Regarding the number of the blockchains, the authors in [37] employed a hierarchical blockchain architecture to ensure good scalability, high throughput, and less transaction latency. In [52], Hussein et

al. used two blockchains, namely policy chain and data chain. Conclusively, it is safe to estimate that the choice of the type of blockchain is related to the nature of the stored data. Furthermore, using two different blockchains one for data and one for policy, can accelerate the search operation and improve the blockchain management.

**Table 3.** Cr1 evaluation.

| Reference | Type of Blockchain | Number |
|---|---|---|
| [27] | Private | One blockchain |
| [34] | Private | One blockchain |
| [36] | Private | One blockchain |
| [37] | Consortium | Multiple blockchains |
| [41] | Private | One blockchain |
| [42] | Consortium | One blockchain |
| [44] | Private | One blockchain |
| [49] | Private | One blockchain |
| [52] | Private | Two blockchains |
| [57] | Consortium | One blockchain |
| [58] | Consortium | One blockchain |
| [61] | Consortium | One blockchain |
| [68] | Consortium | One blockchain |

2.    Cr2: type of consensus algorithm used in the blockchain

Several existing frameworks have adopted different consensus algorithms to validate blocks (summarized in Table 4). In [27], Sun et al. used the distributed proof of stake (DPOS) algorithm, which guarantees high levels of scalability. Above that, it offers a fast "delegated" voting system. Unfortunately, the DPOS algorithm can expose the blockchain to issues related to the voting approach. For instance, DPoS users with small stakes may decide that their vote has no significance compared to the votes of larger stakeholders.

**Table 4.** Cr2 and Cr3 evaluation.

| Reference | Transaction/S.Contract | Contacts Number | Consensus Algorithm |
|---|---|---|---|
| [27] | Smart contract | Multiples | DPOS |
| [34] | Smart contract | One contract | NM |
| [36] | Transaction | NA | Lightweight mechanism |
| [37] | Smart contract | Multiple | PBFT |
| [40] | Smart contract | Multiple | PoW |
| [41] | Smart contract | One contract | Endorsement policy |
| [42] | Smart contract | One contract | Endorsement policy |
| [43] | Smart contract | Multiple | PoW |
| [44] | Transaction | NA | PBFT |
| [49] | Smart contract | One contract | Low overhead approach |
| [52] | Transaction | NA | PoA |
| [48] | Smart contract | Multiple | PBFT |

NA: not applicable; NM: not mentioned.

In [40,43], the authors used the proof of work (PoW) algorithm despite the fact that there is the strong requirement of a large computational capacity leading to large consumption of energy. In [37,44,48], the practical Byzantine fault tolerance (PBFT) algorithm was used. This algorithm offers high processing transactions with low latency. In [52], Hussein et al. used the proof of authority (PoA) algorithm that can be considered as a recent family of Byzantine fault tolerant (BFT) consensus algorithms that work on private blockchain. PoA is known as a lighter message exchange algorithm because it improves the performance and the scalability of the system compared to traditional practical Byzantine fault tolerance (PBFT).

In [41], Islam and Madria did not use one of the familiar consensus algorithms, yet there was a deployment of an endorsement policy wherein the data owner specifies the identity of all endorsing peers using a configuration transaction. Therefore, these findings can help judge that choosing an efficient consensus algorithm is an issue that needs to be addressed. For this purpose, several criteria must be evaluated, namely energy consumption, computing capacity, scaling, and latency.

3.  Cr3: smart contract and number of contracts

After a deep analysis of the proposed solutions (summarized in Table 4), it is clear that some authors only used transactions in the blockchain [36,44,52]. These transactions can be of different types according to their functions such as access transaction, update transaction, and add transaction. In [52], Hussein et al. used two blockchains to reduce the transaction's research time. For this purpose, they used two different types of transactions, namely policy transactions and data transactions. Some others combined smart contract technology with transactions to achieve an effective blockchain-based access control system. The difference in these solutions depends on the distinction of smart contracts number.

In [34], Novo used one smart contract in which all the operations allowed in the system were defined. These operations are triggered by blockchain transactions. This solution is problematic for the reason that the manager who controls the contract can be a malicious user. In [41], Islam and Madria used a single smart contract where they implemented the policy evaluation. They divided security policies into two categories, meta-policies which are immutable, having the power to define who can modify or delete a security policy. In addition, the authors defined security policies, which are based on the ABAC model. These policies are stored in the blockchain using transactions. In [42], Zhang et al. proposed a system where a chain code is used on authority nodes. This solution consists of transactions being used to invoke the chain code to record the access information and transfer the access control decision to the requester. In [46], Zhai et al. deployed one smart contract on the blockchain to improve decision-making efficiency. In [56], Dukkipati and Zhang deployed one smart contract to verify the user's policy but they used an external database system to store security policies. These solutions can lead to policy manipulation attacks. In [27,37,40,48], several smart contracts were used.

In conclusion, it can be notable that smart contracts are faster, simpler, and have reduced system administration. Although they have several advantages, they are still prone to problems. For instance, an error in the code can be expensive to correct and consume a lot of time. Hence, to propose a blockchain-based access control, there is a high chance of confronting certain issues, such as choosing to use an intelligent contract with transactions or only limiting ourselves to transactions. In addition, it is necessary to know the number and the content of these smart contracts.

4.  Cr4: the blockchain platform

Blockchain technology can be used in different ways. The first implemented blockchain was Bitcoin [30]. It used several features such as cryptography, peer-to-peer network, and the PoW as a consensus algorithm. In 2013, a Bitcoin developer named Vitalik Buterin built the Ethereum platform [70], where the main goal was to make the development of decentralized applications easier. The Ethereum platform uses Solidity language to implement smart contracts. To take advantage of the robustness of the Bitcoin network, a

new open source platform has been developed; this is the Rootstock blockchain. The latter uses the merge mining principle.

On the other hand, the Linux foundation proposes the hyperledger blockchain [70]. Several projects are handled such as hyperledger fabric and hyperledger caliper. The first one is used to implement applications and solutions with modular architecture. Additionally, the hyperledger caliper permits the evaluation of the performance of a blockchain implementation using some predefined use cases. Table 5 summarized the different works with their blockchain platforms. It can be stated the Ethereum platform is the most used one.

**Table 5.** Cr4 evaluation.

| Reference | Platform |
|---|---|
| [34] | Ethereum |
| [35] | Ethereum |
| [37] | Hyperledger Fabric |
| [39] | Ethereum |
| [40] | Ethereum |
| [41] | Hyperledger Fabric |
| [43] | Ethereum |
| [48] | Hyperledger Fabric |

5.    Cr5: Access control nature:

In Section 4, a brief description of the recent blockchain-based authorization solutions is provided. These solutions were classified according to their access control nature, namely fully decentralized and partially decentralized. Table 6 summarizes these works. Significantly, the fully decentralized category encompasses token-based, policy-based, cryptography-based, and the hybrid-based architectures. The hybrid solutions combined the        policy-based        architecture        and        the        cryptography-based        one. It can also combine token-based and policy-based architectures.

Apparently, in [48], Ali et al. combined the policy-based architecture with a permission delegation mechanism. This last solution is considered as fully decentralized authorization due to several entities participating in the authorization steps, namely policy creation, policy validation, and permission delegation. Moreover, the partially decentralized category includes the token-based, policy-based, and token and cryptography-based architectures. Hence, in essence, ensuring a fully decentralized access control solution can require guaranteeing decentralization at all phases of the authorization step, regardless of the chosen architecture.

6.    Cr6: Access control phases

Some authors proposed solutions taking into consideration both phases in parallel. Others, however, were limited to a single phase. In [50], Vangala et al. only focused on the authentication phase in the smart farming environment without managing the authorization step, and merely using the elliptic curve cryptography (ECC) to ensure authentication at two levels: device-to-device (D2D) authentication and device-to-gateway (D2G) authentication.

In [35,39,44,46,57,62], the authors proposed schemes that manage the authentication and the authorization phases at the same time. Both steps are deemed to be important to ensure system security. Proposing a solution that considers both steps at the same time is a positive point. However, this issue is difficult to manage because it is necessary to know how to combine the most effective methods of the two phases while guaranteeing the security requirements of a distributed and large-scale environment such as the IoT.

**Table 6.** Cr5 evaluation.

| Reference | AC Architecture | AC Nature |
|---|---|---|
| [27] | Policy based | Partially D |
| [34] | Policy based | Partially D |
| [35] | Token based | Partially D |
| [36] | Policy based | Partially D |
| [37] | Policy based | Partially D |
| [38] | Token and cryptography based | Partially D |
| [57] | Policy and token based | Fully D |
| [40] | Policy based | Fully D |
| [41] | Policy based | Fully D |
| [42] | Policy based | Fully D |
| [43] | Token based | Fully D |
| [44] | Cryptography based | Fully D |
| [46] | Policy based | Fully D |
| [47] | Cryptography and policy based | Fully D |
| [48] | Policy and Permission delegation based | Fully D |

## 7. Cr7: Access control models

A policy-based authorization and access control model is necessary to encapsulate security policies. Analyzing the policy-based authorization solutions presented recently (summarized in Table 7) showed that a large number of these solutions adopted the ABAC model [37,41,42,46,56,68,69]. Although the ABAC model is more flexible and scalable than other access control models, a few disadvantages in a dynamic environment require real-time access control such as IoT. To overcome the issues that an ABAC model can face in this type of environment, an efficient multi-Level security attribute-based access control scheme was presented in [71]. Additionally, the RBAC model was adopted in [27], where the scheme builds a user role table to locate users. Following this procedure, the authors were able to extend the general RBAC model to be able to provide secure and fine-grained access control.

**Table 7.** Cr7 evaluation.

| Reference | Access Control Model |
|---|---|
| [27] | RBAC |
| [37] | ABAC |
| [41] | ABAC |
| [42] | ABAC |
| [46] | ABAC |
| [56] | ABAC |
| [68] | ABAC |
| [69] | ABAC |
| [71] | MLS-ABAC |
| [72] | HABAC |

In [72], Ameer et al. noticed that each IoT application domain has challenges to consider when choosing the access control model. For this reason, the HABAC model was proposed, which was an attribute-based access control model which was especially designed for a smart home context. To conclude, the choice of an adequate access control model remains an issue to be addressed since it depends on several factors such as the application's domain, the nature of the environment, and its security requirements.

## 8. Cr8: Domain application

In Section 5, the existing blockchain-based access control solutions were classified according to the domain applications, namely: smart farming, smart health, intelligent transportation systems, smart grids, and Industry 4.0. A deep analysis of the recent solutions, reveals that access control in smart farming has not been widely considered by current researchers. In fact, two related works have recently been proposed. Consequently, there are insufficient research resources. Moreover, the authors in [49] proposed a blockchain-based solution managed by an administrator. This last point allows us to estimate that it is a solution close to traditional centralized access control approaches. Furthermore, Vangala et al. in [50] focused only on the authentication phase. In summary, as mentioned above, this field of application requires further in-depth studies on its security needs to move towards an adequate access control solution.

Regarding the E-health applications, it cannot be denied that several works have been proposed in this field. Indeed, a recent survey was presented in [51], wherein Sookhak et al. proposed a taxonomy of the different blockchain-based access control solutions. Furthermore, it is safe to assert that the attention for this domain application can also be due to the health situation that the world has been experiencing in the last three years. Additionally, it may be due to the sensitive nature of medical data as malicious access can cost human lives.

Concerning smart transportation systems, most blockchain-based access control solutions targeted the smart parking issue. Dukkipati and Zhang in [56] proposed a blockchain-based access control model in which they tried to minimize the number of security policies. To validate their solution, they took into consideration the scenario of a security policy that permits the sharing of information about parking slots, traffic signals flow between two signals. Furthermore, in [57,58], the authors focused on the smart parking scenario.

After analyzing smart grids access control schemes, it is notable that the cryptography is largely used [59–62]. In [60,61], the authors proposed a cryptography- and policy-based solution. In [62], Nasser et al. used elliptic curve cryptography (ECC).

Finally, regarding Industry 4.0 applications, different access control solutions were proposed. In [68,69], the authors presented a policy-based access control. In both solutions, the scheme was based on the ABAC model and used multiple smart contracts. In [63], Wan et al. used the ECC to ensure data privacy. In [67], Lahbib et al. intended to guarantee the requester's anonymity. For this purpose, they used ring signatures. Therefore, it is possible to say that each IoT domain has its relative security requirements. These must later be identified before designing the authorization solution.

## 9. Cr9: Implementation of the solution and evaluation criteria

In [34], Novo implemented a proof of concept (PoC) prototype of the proposed solution evaluating the influence of the new management hub on system scalability and performance. The author used the Ethereum blockchain with a single smart contract implemented with solidity language. The latency of access control operations was also evaluated, and the throughput in the management hub using different scenarios.

In [36], Algarni et al. did not implement the proposed solution, they left this step for future work. There is also an intention to resolve the big header size issue. In [37], Abdi et al. performed simulations of the proposed solution by using the Hyperledger Fabric blockchain platform. Additionally, the Golang language was used to implement smart contracts. The authors used the Hyperledger Caliper to evaluate the performance of their solution. For this purpose, they calculated the transaction latency

and transaction throughput. In [39], Xu et al. implemented a proof of concept prototype of the proposed scheme BlendCAC. The Ethereum blockchain with Solidity language was employed in this procedure to implement the smart contract. As hardware equipment, the authors adapted two Raspberry Pi 3, two laptops, and four desktops. The mining process was performed by laptops and desktops while the two raspberry Pi 3 played the role of client and service provider, respectively. Furthermore, the authors evaluated the computational and communication overheads of the proposed solution. As a result, the BlendCAC scheme seems to have less processing time than the RBAC and ABAC models. Additionally, a small amount of overhead was introduced by the BlendCAC scheme. The authors believe that it can be improved if the scheme will be implemented on more powerful smart devices.

In [40], Zhang et al. provided a case study to show the application of the proposed solution in the IoT. As hardware, they used one desktop computer, one laptop, and two Raspberry Pi 3 Model B. Furthermore, the procedure consisted of employing the Ethereum blockchain and the Solidity programming language to implement smart contracts. The number of gas required to deploy the three smart contracts was evaluated. Parenthetically, the gas is a unit used in the Ethereum platform to measure the capital cost to execute a smart contract. Furthermore, the authors calculated the average time required to deploy the three smart contracts.

In [41], the proposed access control system was fully implemented. First, Islam and Madria developed an IoT test-bed. Afterward, they implemented the blockchain network in Hyperledger Fabric v1.3. The evaluation of the proposed solution shows that it can treat access control requests of IoT resources faster than the public blockchain and that using the optimum parameter values (block size: 20 and 40 transactions per second, block timeout = 1 s).

In [43], Nakamura et al. implemented the capability, delegation graph, and the token's creation, delegation, revocation, and verification functions. They used one MacBook Pro, one MacBook Air, and two Raspberry pi as hardware. The evaluation of the proposed solution is based on a private Ethereum blockchain. Nakamura et al. evaluated the gas consumption of the proposed scheme in the case of token creation, token delegation, and token revocation. Furthermore, they compared the obtained results to those obtained by the BlendCAC scheme [39]. The experimental results have shown that the proposed solution needs less gas than the BlendCAC scheme. In conclusion, the implementation of the proposed solution can be estimated as the unique point that allows us to validate the proposal and this is after the definition of the most important criteria that need to be evaluated.

This section is dedicated to a comparison between this work and the studies previously summarized in Section 2. This comparison is based on some criteria deemed important (illustrated in Table 8) . In addition, an in-depth analysis of recent blockchain-based access control solutions (carried out in Section 6) also consisted of using the same criteria. Table 2 illustrates a brief description of each criterion. In [15], Maw et al. satisfied the Cr5 and the Cr9 criteria while Rouhani and Deters in [17] took into consideration the Cr3, Cr5, and the Cr8 criteria. Moreover, it is shown that in [16,18,19] authors took into account only one criterion: Cr5, Cr3, and Cr1, respectively. Furthermore, Hussein et al. [20] managed Cr3 and Cr9 criteria. Furthermore, it can be noticed that none of the previously cited comparison criteria was managed by Shantanu et al. [21]. Finally, According to this analysis, it is clear that only our work took into account all these comparison criteria.

**Table 8.** Related works comparison.

| References | BC Features | | | | AC Features | | | G. Features | |
|---|---|---|---|---|---|---|---|---|---|
| | **CR1** | **CR2** | **CR3** | **CR4** | **CR5** | **CR6** | **CR7** | **CR8** | **CR9** |
| [15] | X | X | X | X | ✓ | X | X | X | ✓ |
| [16] | X | X | X | X | ✓ | X | X | X | X |
| [17] | X | X | ✓ | X | ✓ | X | X | ✓ | X |
| [18] | X | X | ✓ | X | X | X | X | X | X |
| [19] | ✓ | X | X | X | X | X | X | X | X |
| [20] | X | X | ✓ | X | X | X | X | X | ✓ |
| [21] | X | X | X | X | X | X | X | X | X |
| Our work | ✓ | ✓ | ✓ | ✓ | ✓ | ✓ | ✓ | ✓ | ✓ |

X: Not supported; ✓: supported.

This section includes an analysis of the recent blockchain-based access control models in the IoT environment. This analysis is based on three categories of criteria deemed important. The first group deals with the criteria relative to the blockchain technology whereas the second group encompasses the criteria that are concerned with the access control mechanism. The last group is defined as general feature criteria such as domain application and implementation. Adding on that, this paper provided a comparison between the presented content and the related work papers, and this comparison showed that only this work took into consideration the three categories of criteria.

## 7. Discussion and Open Issues

The analysis of the existing blockchain-based access control solutions allowed the authors to deduce that proposing a blockchain-based authorization technique faces several issues. The latter can be classified into two groups. The first group deals with the issues that are related to access control in a distributed and scalable environment such as IoT, whereas the second group encompasses issues related to blockchain technology. These challenges are explained in detail in the following.

1. Access control-related issues

    There are some issues with the access control such as:

- How to propose a decentralized and lightweight access control solution for distributed and high-scale IoT environments even though IoT devices have limited storage and computing capacity.
- Should the proposed solution combine the two access control phases or does it just take into consideration one phase? Furthermore, which technique can be chosen to ensure an effective authorization (fully decentralized, partially decentralized)?
- What is the suitable access control model used in the proposed access control solution? Will an existing model be used or will a combination of several models be necessary? What are the criteria that can be used to determine an appropriate access control model?

2. Blockchain-related issues

    Blockchain technology represents an appealing solution to ensure access control decentralization. However, blockchain-enabled access control is a very promising scheme and permits dealing with a single point of failure problems. However, there are still some potential problems related to the blockchain technology that need to be addressed. Some of these issues have already been discussed in [33].

- The first issue is the harmonization (interoperability) of blockchain across application areas and geographic locations.
- The second issue is the power consumption associated with the consensus algorithm that solves the double-spending problem.

- The third issue is the bandwidth overheads associated with the blockchain which can increase the latency.
- The last issue concerns the implementation of the blockchain algorithm on diminutive devices.

Several solutions can be proposed to overcome the issues mentioned above. In a blockchain, there is an obligation on each node to perform a comparable task for the verification of each transaction at the same time. This comparison generates a high computation cost. Gupta et al. [73] proposed a game theory-based authentication framework with blockchain technology to resolve the Internet of Vehicles (IoV) cross trusted authority's authentication issues. To manage the network scarcity challenges, Islam et al. [38] proposed a lightweight scheme employing drones to assist IoT devices for secure data collection. Furthermore, dew computing was used to permit offline computations. Kumar et al. [74] proposed a blockchain-based edge framework (BlockEdge) to ensure low-latency services for IIoT applications, while Sosu et al. [75] presented a review paper where they highlighted the importance of integrating blockchain technology with dew computing. Furthermore, machine learning can be suggested as a solution to some of the previously discussed challenges. Xiao et al. [76] showed how machine learning techniques can help to build lightweight access control protocols in a heterogeneous environment with multi-source data and multiple types of nodes such as IoT. To solve the complex computation offloading of vehicles while ensuring the high security of the cloud server in vehicular ad hoc networks (VANETs), a blockchain-based solution was proposed in Zheng et al. [77]. This solution considers offloading tasks by optimizing offloading decisions, consensus mechanism decisions, the allocation of computation resources, and channel bandwidth. The access control model issue was managed in [78] where Zhaou et al. applied machine learning to obtain a fully user-role assignment process in the RBAC model in the SCADA system. The combination of a blockchain and physically unclonable function for IoT is an interesting area of research [79] to solve bandwidth, integration, scalability, latency, and energy requirements for the Internet of Energy (IoE) systems.

## 8. Conclusions

This article first explained the importance of using blockchain technology in access control in an IoT environment. Next, a classification of recent works on blockchain-based access control was presented according to the nature of the approach: fully decentralized or partially decentralized. Additionally, the recent solutions were classified and analyzed according to the IoT applications. Finally, a deep analysis of recent authorization frameworks was performed according to three categories of criteria. The first one concerns blockchain technology. The second category of criteria is relative to access control. The last group represents general criteria. In our future works, we plan to present a survey on blockchain-based authentication schemes. One of the obvious concerns that will be discussed is whether these solutions can follow the same taxonomy proposed in this paper. Since these schemes are based on blockchain technology, there must be a notion of decentralization. It remains to be seen whether this decentralization is total or partial. Adding to that, an analysis of the relationship between the authentication technique and authorization technique will be provided. Furthermore, an important question that cannot be neglected is whether the choice of authentication technique influences that of authorization, namely: policy-based, token-based, and cryptography-based. Additionally, there is a plan to study several paradigms such as fog computing, multi-access edge computing, and dew computing to see whether they permit us to implement blockchain-based access control techniques to reduce the computation cost.

**Author Contributions:** S.N. and I.B.D. equally contributed to the manuscript concept, methodology, original draft writing, visualization and editing. I.B.D. secured the funding of the manuscript. All authors have read and agreed to the published version of the manuscript.

**Funding:** This research received no external funding.

**Conflicts of Interest:** The authors declare no conflict of interest.

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
