# Peer review of "Blockchain-Based Access Control Techniques for IoT Applications"

_electronics, doi:10.3390/electronics11142225_

Round 1

Reviewer 1 Report

Authors proposed a blockchain-based access control taxonomy according to the access control nature: partially decentralized and fully decentralized. Furthermore, it presents an overview of blockchain-based access control solutions proposed in different IoT applications. 

The manuscript is well written and can be accepted for publications after incorporating the following changes. 

1. How can you conclude that "Security of the IoT ecosystem has been identified as a major bottleneck" ? Provide evidence.

2. The motivation of the work is not properly outlined in section 1. Paragraph before contribution para needs modification.

3. Authors must highlight some attacks in IoT in section 1 to lay down proper foundation. The following work will be useful. Information Security threats and attacks with conceivable counteraction; Classification and analysis of security attacks in WSNs and IEEE 802.15. 4 standards: A survey.

4. Section 2 can be summarized in the form of a table. Authors must highlight and compare the contribution of various previous works that have been discussed.

5. Link between sections and subsections are needed. Check section 3 and 3.1.

6. Section 4.1 looks weak and is unacceptable. Blockchain has immense potential for IoT applications and the same must be presented properly. Refer to the work titled Untangling blockchain technology: A survey on state of the art, security threats, privacy services, applications and future research directions

7. Authors fail to elaborate BIoT Applications. The following works may be useful. Blockchain based solutions to secure IoT: background, integration trends and a way forward; Unifying blockchian and IoT: security requirements, challenges, applications and future trends.

8. More future research directions must be outlined to guide further research in the area.

Author Response

Dear Reviewer,

Coauthor and I are indebted to your pertinent comments and suggestions that helped improve the content and scope of the manuscript. We have carefully read all your valuable comments and addressed them in the revised edition. Below are the list changes in light of your comments/ suggestions. The edited part of the manuscript is highlighted in the revised document.

Reviewer 2 Report

1. related work should be consider recently also , existing work limitations did not mention clearly.

2. figure 1 and 2 need more detail explanation.

3. problem and challenges did not well discuss also what are the possible solutions.

4. figure 6 , how can these techniques are use and which technique is effective?

5. future work need to explain clearly.

Author Response

(The authors gave the same response as above.)

Reviewer 3 Report

The authors discussed a blockchain-based access control survey considering partially decentralized and fully decentralized along with blockchain-based access control solutions proposed in different IoT applications. The idea seems interesting. However, I have the following concerns. 

1. Please revise the grammatical errors of this paper.

2. Recent blockchain access control technique-related papers are missing. Some are mentioned as follows.

-> A survey on Blockchain based access control for Internet of Things," 2019 15th International Wireless Communications & Mobile Computing Conference (IWCMC), 2019, pp. 502-507, doi: 10.1109/IWCMC.2019.8766453.

-> "FBI: A Federated Learning-Based Blockchain-Embedded Data Accumulation Scheme Using Drones for Internet of Things," in IEEE Wireless Communications Letters, vol. 11, no. 5, pp. 972-976, May 2022, doi: 10.1109/LWC.2022.3151873.

3. Survey regarding blockchain-based access control is a very popular topic. Highlight your novelty. 

4. add a summary table for Section 2 along with their limitation.

5. Revise Section 7 and put open issues in bullet form for better readability. 

6. The authors put more priority on Fog computing. It would be better author can highlight other technology like multi-access edge computing, dew computing, and so on. 

Author Response

Dear Reviewer,

The coauthor and I are indebted to your pertinent comments and suggestions that helped improve the content and scope of the manuscript. We have carefully read all your valuable comments and addressed them in the revised edition. Below are the list changes in light of your comments/ suggestions. The edited part of the manuscript is highlighted in the revised document.

Round 2

Reviewer 2 Report

All comments are well addresses.

Author Response

Coauthor and I are indebted to your pertinent comments and suggestions that helped improving the content and scope of the manuscript. We  are happy and glad that we are satisfied with the way we addressed your valuable comments. We have performed a proof reading of the manuscript using a professional tool. The modifications are highlighted in the revised manuscript.

Reviewer 3 Report

The authors have revised their paper. However, they didn't revise the paper carefully. 

1. Please revise the grammatical errors of this paper. 

Response- Thank you so much for your suggestion. Grammatical errors have been fixed.

Outcome: Still, there are a lot of errors exist throughout the paper. For example, 

On page 1, Line 27, the Authors wrote: 

"The network layer responsible for connecting the perception layer to the global internet"

Which will be 

"The network layer is responsible for connecting the perception layer to the global internet"

2. Recent blockchain access control technique-related papers are missing.

Response: Regarding the second paper you give us, authors proposed a federated learning-based data accumulation scheme that combines drones and blockchain technology. The proposed solution contains a two-phase authentication mechanism. Authors neglected the authorization phase. The taxonomy that we presented in section 4.3 concerns the authorization phase. Besides, a classification of the existing blockchain based access control solutions according to IoT applications is presented in section 5. The paper you give us is very interesting but it did not mention application domain of the IoT. It is only for the reasons previously mentioned that we have not added it to the manuscript.

Comment:  I think the authors didn't read the paper carefully. "Regarding the second paper you give us, authors proposed a federated learning-based data accumulation scheme that combines drones and blockchain technology" -> data that authors mentioned comes from internet of things data and it also gets validation during sending requests. Thus, this "The paper you give us is very interesting but it did not mention application domain of the IoT." reason doesn't seem valid. Moreover, "Blockchain-based Access Control Techniques for IoT Applications" -> paper focused both on authentication and authorization (as shown in Fig. 6). Then, why authors looked for comparing in the authorization part instead of authentication? The author needs to revise their analysis.  

3. Instead of writing as [13], try to write in "Author et al [13]" format throughout the paper. 

4. Comment: The authors put more priority on Fog computing. It would be better author can highlight other technology like multi-access edge computing, dew computing, and so on.

Answer: Modifications have been done in the conclusion section.

In Conclusion: 

Additionally, there is a plan to study several paradigms such as fog computing, multi access edge computing and dew computing to see if they permit us to implement blockchain based access control techniques reducing the computation cost

Outcome: Authors already discussed fog computing in Section 1, Paragraph 2. Then again put fog computing as a future work in the conclusion. My concern was instead of focusing only on Fog computing (as discussed in Section 1, Paragraph 2), try to be general and focus on all others also for better readability instead of the perspective of implementing blockchain in fog computing, multi access edge computing and dew computing.  

Author Response

Dear Reviewer,

Coauthor and I are indebted to your pertinent comments and suggestions that helped improving the content and scope of the manuscript. We have carefully read all your valuable comments and addressed them in the revised edition. Below are the list changes in light of your comments/ suggestions. Edited part of the manuscript is highlighted in the revised document.

Round 3

Reviewer 3 Report

I am recommending to accept this paper.